# Effect of dietary protein levels on the growth, enzyme activity, and immunological status of *Culter mongolicus* fingerlings

Jing Qian[1,2], Lingjun Xiao[1,2], Kai Feng[1,2], Wei Li[1,3], Chuansong Liao[1,3], Tanglin Zhang[1,2,3], Jiashou Liu[1,2,3]*

**1** State Key Laboratory of Freshwater Ecology and Biotechnology, Institute of Hydrobiology, Chinese Academy of Sciences, Wuhan, Hubei Province, China, **2** University of Chinese Academy of Sciences, Beijing, PR China, **3** National Research Centre for Freshwater Fisheries Engineering, Wuhan, PR China

* jsliu@ihb.ac.cn

**Data Availability Statement:** All relevant data are within the paper and its Supporting Information files.

## Abstract

A 65-day growth trial was conducted to investigate the dietary protein requirements for *Culter mongolicus* fingerlings. Isolipidic and isoenergetic diets were formulated with five dietary protein levels (32%, 37%, 42%, 47%, and 52%). Each diet was assigned to triplicate groups of 70 *C. mongolicus* fingerlings (0.99±0.08 g). The results indicated that weight gain and specific growth rate (SGR) increased with increasing dietary protein levels up to 47%. The activities of intestinal trypsin and lipase were the lowest in the 32% protein and 52% protein groups, while amylase activity reduced markedly in the 47% protein group. These results suggest that different dietary protein levels may cause different transformations of nutrients. The activities of superoxide dismutase (SOD) and lysozyme were not affected by varying dietary protein levels, except for those in the 32% protein group. In contrast, the content of malondialdehyde (MDA) increased with increasing dietary protein levels and reaching a maximum in the 52% protein group, suggesting that MDA accumulation depends on the protein concentration and the potential oxidative stress. Taken together, based on the broken-line analysis of SGR, we recommended the optimum dietary protein for *C. mongolicus* fingerlings to be 48.97%~49.31%.

## Introduction

Deficiencies or excesses of the main dietary components have profound effects on the growth and survival of fish [1]. Protein accounts for 65%~75% of fish dry-weight, and protein deposition in fish appears to be the main determinant of weight gain [2]. It plays a key role in many biological functions, including structural, enzymatic, transport, immune, and cell signaling [3]. Protein is broken down into smaller molecules in the gastrointestinal tract, which secrete fluids, electrolytes, and digestive enzymes that allow the absorption and utilization of free amino acids [4]. Dietary protein deficiency results in retarded growth and poor health [5–8]. On the other hand, excessive dietary protein can promote ammonia excretion and increase utilization of energy for amino acid catabolism, leading to water pollution and retarded growth

**Funding:** This work was financially supported by the National R & D Supporting Program (2015BAD25B01), Fund of the State Key Laboratory of Freshwater Ecology and Biotechnology and the earmarked fund for China Agriculture Research System (CARS-45), and the Shandong Provincial Agricultural Seed Engineering Project (No. 2017LZN003). The contribution of programs are as follows: The National R & D Supporting Program (2015BAD25B01) played a role in study design, the Fund of the State Key Laboratory of Freshwater Ecology and Biotechnology and the earmarked fund for China Agriculture Research System (CARS-45) played a role in data collection and analysis, and the Shandong Provincial Agricultural Seed Engineering Project (No. 2017LZN003) played a role in experiment conduction and data analysis.

**Competing interests:** The authors have declared that no competing interests exist.

[9, 10]. Therefore, the optimum dietary protein level is a key factor for fish to reach optimum growth at a low energy cost.

The dietary protein requirements depend on fish species, and the optimum protein levels differ at different growth stages and/or body sizes [2, 11, 12]. Many studies have focused on the juvenile and broodstock stages, which are better adapted to pellets than that at the younger stages [6, 13–21]. In fact, during larval rearing, a powder-formulated diet could be added as a nutritional supplement as live prey, such as cladocera and copepods, because of the decreasing of live prey amount in pond. As larvae grow, the ingredients and particle sizes of the formulated diet can be adjusted according to the nutritional requirements and size of the fish oral gape, which is a common way to achieve weaning on artificial diets. For most fish species, their developments are far from perfect during the earlier period of domestication, and therefore the protein level of the formulated diet during this period is important for fish somatic growth and development [22, 23].

Digestive enzymes are important for nutrient digestion. Higher enzyme activity in the digestive tract enhances the digestive capability and growth performance of the host. It is widely accepted that the level of fish digestive enzyme activity is a useful comparative indicator of food utilization, digestive capacity, and growth performance of the host [24, 25]. Many factors such as the stage of life, diet, feeding management, and the sampling time after feeding affect the activity of digestive enzymes [26]. Changes in diet composition can modulate enzymatic activities and nutrient absorption capacity, leading to improve feed use and assure growth performance [27].

Immunological status is an important health indicator of an organism. Lysozyme is an important hydrolytic enzyme of the non-specific immune system, and it can disrupt b-(1, 4) glycosidic bonds between the N-acetylmuramic acid and N-acetylglucosamine in the peptidoglycan of bacterial cell walls [28]. As a humoral element in the innate immune system, lysozyme levels can be elevated in response to the immunostimulants, vaccines, and probiotics [29–32]. Superoxide dismutase (SOD) is an important enzyme in the cellular antioxidant enzymatic system, and it can remove the internal reactive oxygen species (ROS), that are generated during the immune response and metabolic process, prevent the occurrence of fatty acid oxidation, decrease the toxic effects of ROS, and consequently protect organisms from oxidative damage [33–35]. Malondialdehyde (MDA) is an indicator of endogenous oxidative damage and a final product of lipid peroxidation [36, 37].

*Culter mongolicus* is widely distributed in freshwater bodies including lakes, rivers, and reservoirs of East Asia [38] and is one of important commercially carnivorous species in China due to the increasing production [39]. However, its population has declined gradually in recent decades, due to water pollution, habitat destruction, and overfishing [40, 41]. This species has been stocked into natural waters such as the Yangtze River basin to conduct biomanipulation [42–47]. Stock enhancement programs have been launched by China's Ministry of Agriculture since the early 2000s to rehabilitate this valuable fishery resource [48]. Hatchery-reared fingerlings were released annually to build the recruitment stock. In addition, larger size fingerlings would improve the recapture rate [49]. Based on earlier research, artificial propagation of *C. mongolicus* was successful [50]. However, the survival rate of fingerlings was lower than 45% due to the shortage of suitable food for their small oral gape. Therefore, it is crucial to develop diets with optimum protein levels to improve the survival rate and enhance the health of *C. mongolicus* during fingerling stage.

To this end, the present study aimed to evaluate the effect of varying dietary protein levels on the growth and the survival rate of *C. mongolicus* fingerlings in artificial cultivation. In addition, we measured the activities of intestinal digestive enzymes and non-specific

immunological indices in the plasma to evaluate the physiological effects of different dietary protein levels.

## Materials and methods

### Ethic statement

*C. Mongolicus* is not an endangered or protected species. All experimental animal care protocols were approved by the ethics committee of the Institute of Hydrobiology, Chinese Academy of Sciences. And all experimental methods were performed following the guidelines for the care and use of experimental animals of China (GB/T35892 2018) [51].

### Experimental diets

Five isolipidic (mean lipid percentage of 7.26%) and isoenergetic (mean 18.25 kJ/kg) experimental diets containing 32%, 37%, 42%, 47%, and 52% crude protein levels (dry basis) were formulated. The formulation and chemical composition of the diets are presented in Table 1. White fishmeal was used as an animal protein source, soybean meal, rapeseed meal, and flour were used as plant protein sources, and fish oil and soybean oil (1:1, w/w) as lipid sources. Corn starch and cellulose were added to the diets to modulate the total energy. All ingredients were passed through a 375μm sieve before mixture. Diets were prepared in a laboratory extruder (SLP-45, Fishery Mechanical Facility Research Institute, Shanghai, China) to form 1.5-mm pellets. The pellets were oven-dried at 75˚C and stored at −4˚C until use.

**Table 1. Formulation and chemical composition of the experimental diets (% in dry matter basis).**

| Ingredients | Dietary protein levels (%) | | | | |
|---|---|---|---|---|---|
| | **32** | **37** | **42** | **47** | **52** |
| White fishmeal[a] | 27.1 | 34.3 | 41.4 | 48.6 | 55.7 |
| Soybean meal | 13 | 13 | 13 | 13 | 13 |
| Rapeseed meal | 13 | 13 | 13 | 13 | 13 |
| Flour | 8 | 8 | 8 | 8 | 8 |
| Fish oil[b] | 1.98 | 1.62 | 1.27 | 0.94 | 0.6 |
| Soybean oil | 1.98 | 1.62 | 1.27 | 0.94 | 0.6 |
| Corn starch | 27.44 | 20.95 | 14 | 6.8 | 0 |
| Cellulose | 0 | 0.01 | 0.56 | 1.22 | 1.6 |
| Vitamin premix[c] | 0.39 | 0.39 | 0.39 | 0.39 | 0.39 |
| Mineral premix[d] | 5 | 5 | 5 | 5 | 5 |
| Carboxymethyl cellulose | 2 | 2 | 2 | 2 | 2 |
| Choline chloride | 0.11 | 0.11 | 0.11 | 0.11 | 0.11 |
| Chemical composition (% in dry matter) | | | | | |
| Crude protein | 32.02 | 37.04 | 42.00 | 47.03 | 51.98 |
| Crude lipid | 7.27 | 7.25 | 7.24 | 7.27 | 7.28 |
| Gross energy (kJ g$^{-1}$)[e] | 18.18 | 18.27 | 18.26 | 18.25 | 18.28 |

[a] Pollock fishmeal from American Seafood Company, Seattle, Washington, USA.

[b] Anchovy oil from Peru purchased from Coland Feed Co. Ltd., Wuhan, Hubei, China.

[c] Vitamin premix (mg kg$^{-1}$ diet): thiamin, 20; riboflavin, 20; pyridoxine, 20; cyanocobalamine, 0.020; folic acid, 5; calcium pantothenate, 50; inositol, 100; niacin, 100; biotin, 0.1; starch, 645.2; ascorbic acid, 100; vitamin A, 110; vitamin D, 20; vitamin E, 50; and vitamin K, 10.

[d] Mineral premix (mg kg$^{-1}$ diet): NaCl, 500; MgSO$_4$·7H$_2$O, 8155.6; NaH$_2$PO$_4$·2H$_2$O,12500.0; KH$_2$PO$_4$, 16,000.0; CaHPO$_4$·H$_2$O, 7650.6; FeSO$_4$·7H$_2$O, 2286.2; C$_6$H$_{10}$CaO$_6$·5H$_2$O,1750.0; ZnSO$_4$·7H$_2$O, 178.0; MnSO$_4$·H$_2$O, 61.4; CuSO$_4$·5H$_2$O, 15.5; CoSO$_4$·7H$_2$O, 34.5; KI, 114.8; and corn starch, 753.7.

[e] Gross energy obtained through calorimetry.

## Experimental fish and feeding trial

*Culter mongolicus* fingerlings were obtained from the Niushan Lake Fish Farm (Wuhan, Hubei, China) and transported to the laboratory of the Institute of Hydrobiology, the Chinese Academy of Sciences (Wuhan, Hubei, China). Prior to the experiment, all fish were acclimated to the laboratory rearing system. Similar sized fish (mean initial total length 55.08±1.49 mm; mean initial weight 0.99±0.08 g) were randomly distributed into 15 cylindrical tanks (diameter 60 cm, water depth 60 cm). Each diet was assigned to triplicate tanks at a density of 70 fish per tank. Fish were hand-fed their prescribed diets to apparent satiation twice a day at 09:00 and 16:00.

At the beginning of the experiment, the granular feeds were milled and rolled into balls with water to prevail upon fish to ingest formula feeds, and then pellets were added gradually. All fish were fed with the pellets feeds initiatively after 15 days. The uneaten feeds and feces were removed an hour after feeding. A third of the water was replaced at 19:00 daily. A photo-period of 12 h light:12 h dark cycle was maintained (lighting-up time was 8:30~20:30). During the experiment, the averaged water temperature was 25.0°C, with a dissolved oxygen (DO) level >5.0 mg/L, pH 7~7.5, ammonia-N < 0.5 mg/L, and residual chlorine < 0.05 mg/L. The trial lasted for 65 days.

## Sample collection

At the end of the feeding experiment, all fish were starved for 24 h before sampling. Total body length, weight and mortality of fish in each tank were recorded to calculate the specific growth rate of total length ($SGR_{TL}$) and body weight ($SGR_{BW}$), and the survival (SR). Blood samples were collected from the caudal veins (sixty specimens per tank) using heparinized syringes. After centrifugation (3500 rpm, 15 min, 4°C) (centrifuge 5417R, Eppendorf, Hamburg, Germany), plasma was separated and stored at −20°C until analysis of lysozyme, superoxide dismutase (SOD), and malondialdehyde (MDA) concentrations. After blood sampling, the intestine was dissected on ice, and the intestinal samples were frozen immediately in liquid nitrogen and stored at−80°C until analysis of trypsin, lipase, and amylase activities.

## Growth parameters

$$\text{SR}: \text{ Survival rate (\%)} = 100 \times \text{final number of fish/initial number of fish} \qquad (1)$$

$$\begin{aligned}\text{SGR}_{BW}: &\text{ The specific growth rate of body weight (\% d}^{-1}) \\ &= 100 \times \text{Ln (final body weight/initial body weight)/days}\end{aligned} \qquad (2)$$

$$\begin{aligned}\text{SGR}_{TL}: &\text{ The specific growth rate of total length (\% d}^{-1}) \\ &= 100 \times \text{Ln (final total length / initial total length)/days}\end{aligned} \qquad (3)$$

## Chemical analysis of feeds

The chemical composition of the experimental diets was analyzed using the following standard methods. The samples were dried to a constant weight at 105°C for 24 h to determine the dry matter content. Crude protein content was determined by the Kjeldahl method using a Kjeltec system (Kjeltec-8400, FOSS). Crude lipids were measured by Soxhlet extraction using a Soxhlet extractor (Soxtec-2055, FOSS). Crude ash content was determined by incineration in a muffle furnace at 550°C for 12 h. Gross energy was determined using an automatic oxygen bomb calorimeter (Parr Isoperibol Calorimeter 6200, Moline, Illinois, USA).

## Analysis of immunological parameters

Plasma malondialdehyde (MDA) levels and superoxide dismutase (SOD) activities were measured according to the instructions of the commercial assay kits (Nanjing Jiancheng Bioengineering Institute, Nanjing, Jiangsu, China). Lysozyme activity was measured as described by Parry and Ellis [52, 53], based on the lysis of *Micrococcus lysodeikticus* (Sigma Chemical C) with some modifications. Twenty microliters of plasma was mixed with 250 μl *M. lysodeikticus* suspension (0.3 mg/mL in 0.05 M PBS, pH 6.2). The mixture reacted at 25°C for 0.5 min and 4.5 min, and then the optical density (O.D.) was measured at 0.5 min and 4.5 min respectively at 490 nm (Photometer 5010, BM Co. Germany). One unit of enzyme activity was defined as the amount of enzyme causing a decrease in absorbance of 0.001 and the activity was expressed as U/min/mL plasma.

## Quantification of digestive enzymes

Intestinal trypsin, lipase, and amylase were extracted, and their activities were measured according to the instructions of the commercial assay kits (Nanjing Jiancheng Bioengineering Institute, Nanjing, Jiangsu, China). According to the kits instructions, frozen intestinal samples were homogenized in a specific solvent at a propotion of 1:9, followed by centrifugation (2500 g, 10 min, 4°C) to obtain supernatants for testing trypsin activities; frozen intestinal samples homogenized in ice-cold 0.68% saltwater at a proportion of 1:4, followed by centrifugation (2500 g, 10 min, 4°C) to obtain supernatants for testing lipase activities; and frozen intestinal samples homogenized in ice-cold 0.68% saltwater at a proportion of 1:9, followed by centrifugation (2500 g, 10 min, 4°C) to obtain supernatants for testing amylase activities.

## Statistical analysis

All data were subjected to one-way analysis of variance (ANOVA) using SPSS 22.0 (SPSS, IL, USA), after checking for the normality and homogeneity, followed by Duncan's multiple comparison analysis to conduct pairwise comparisons. The optimum dietary protein requirement based on the total length and weight-specific growth rate was estimated using broken-line regression analysis ($y = L - U \times (R - x)$, y means specific growth rate, x means protein content) [17, 54] in Excel 2016. The data were expressed as mean ± SE (standard error).

# Results

## Growth performance

There was no mortality in *C. mongolicus* fingerlings during the feeding trial. The body weight and total length increased with the increasing dietary protein and then significantly decreased at 57% dietary protein. Similarly, $SGR_{TL}$ and $SGR_{BW}$ increased significantly ($p < 0.05$) when dietary protein levels increased from 32% to 47% and then a decreasing trend was observed in the 52% protein group. Based on broken-line regression analysis of $SGR_{TL}$ and $SGR_{BW}$ against dietary protein levels, the optimum dietary protein levels for maximal growth of the fish were 49.31% and 48.97% (Fig 1).

## Digestive enzyme activities

The intestinal digestive enzyme activities in intestine of fish are shown in Fig 2. The activities of intestinal trypsin in fish fed on 37%, 42%, and 47% protein diets were not significantly different ($p > 0.05$), but were significantly higher than those of fish fed on 32% and 52% protein diets ($p < 0.05$). The intestinal lipase activities were significantly lower in the 32% and 52% protein groups than in the other three groups ($p < 0.05$), while no significant difference was

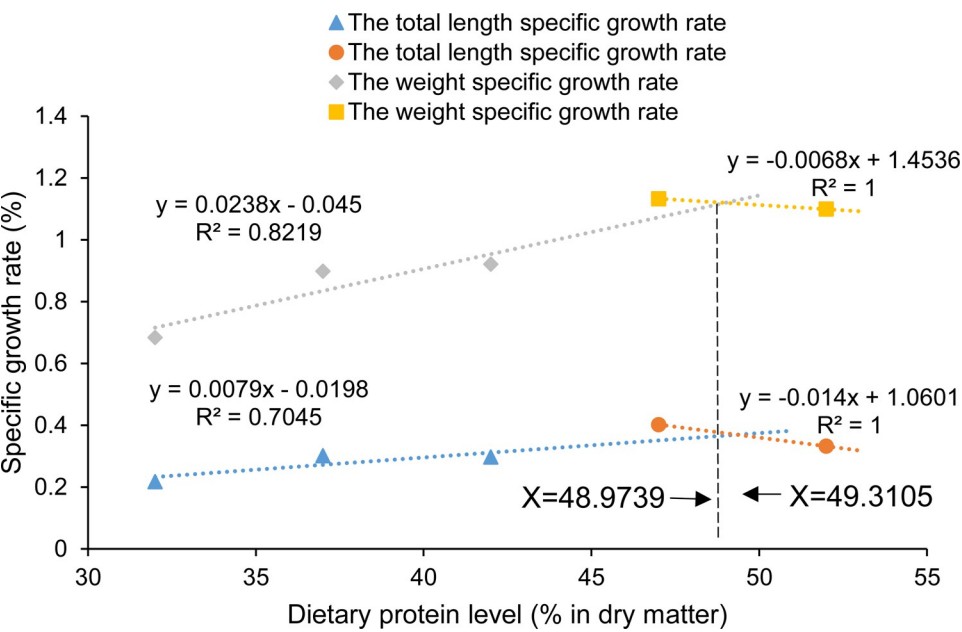

**Fig 1. Relationship between varying dietary protein levels and specific growth rate of *Culter mongolicus*.**

observed among those three groups. The intestinal amylase activity in the 47% protein group was the lowest among the test groups ($p < 0.05$), and no significant difference was observed among the other four groups.

## Immunological parameters

The results of immunological parameters in plasma are shown in Fig 3. The lysozyme activity of fish that were fed the 32% protein diet was significantly higher than that in the other groups ($p < 0.05$), and there was no significant difference among the other groups. MDA content increased with the increasing dietary protein levels ($p < 0.05$). SOD activity did not show any significant difference among the experimental groups ($p > 0.05$).

## Discussion

### Growth performance

The results of this study showed that the 100% survival of *C. mongolicus* fingerlings, indicating an adaptation to a wide range of dietary protein levels. These findings are similar to these of previous studies on Brazilian sardine [8], *Rhamdia quelen* [13], *Nibea diacanthus* juveniles [17], Dabry's sturgeon juveniles [54], bluegill sunfish juveniles [55], and juvenile marbled spinefoot rabbit fish [56]. These results suggest that dietary protein levels may not be a determinant factor of the mortality.

The growth of *C. mongolicus* was profoundly affected by the different dietary protein levels. The values of $SGR_{TL}$ and $SGR_{BW}$ revealed high dietary protein requirements for the growth of *C. mongolicus*. However, excessive protein content may have no further benefits for individual growth and may even induce growth retardation due to the higher energy cost of extra protein metabolism [3, 57, 58]. Similar tendencies were found in several other fish species, such as the Brazilian sardine [8], juvenile bluegill sunfish [55], and juvenile red spotted grouper [59]. The optimum dietary protein requirement for *C. mongolicus* is about 49%. This value was higher

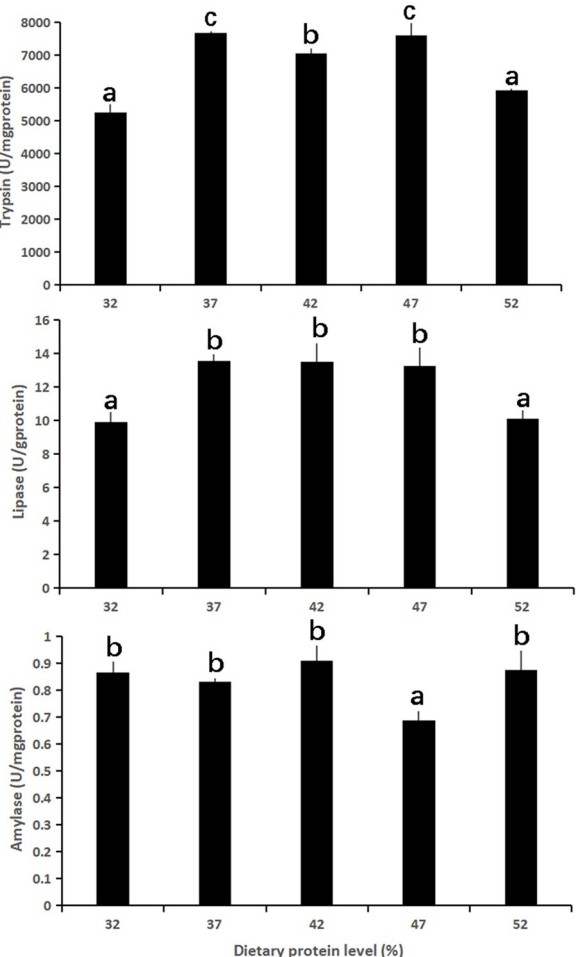

**Fig 2. Specific activities of three digestive enzymes in the intestine of *Culter mongolicus* fed with varying dietary protein levels.** Data presented are mean ± SE, different letters on the top of bars mean significantly different ($p < 0.05$).

than the values reported for the fingerlings of similar sizes, such as those of *Sardinella brasiliensis*, 36.77% [8], *Horabagrus brachysoma*, 39.10% [60] and *Lepomis macrochirus*, 41.51% ~42.37% [55]. However, it is similar to the dietary protein requirements for some commonly cultured carnivorous species, viz., *Salmo salar* (48%) and *Oncorhynchus mykiss* (48%) [3]. The possible reason for this is may be that most herbivorous and omnivorous fish need 250~350 g/ kg of protein in their diets, while carnivorous species require a higher dietary protein contents ranging from 400~550 g/kg [1]. Moreover, the dietary protein requirements generally decrease with fish growth, a phenomenon reported in many fish species at different life stages [3].

## Responses of digestive enzyme activities to dietary protein levels

Several studies have shown that dietary protein levels influence protease activity directly, but the protease activity varies among species. Protease activity in the digestive tract of *Labeo rohita* fingerlings was significantly lower than the optimum level when fed with either insufficient or excessive protein diets [61]. In contrast, the protease activity in *Dentex dentex* [62], *Cyprinus carpio* [63], and *Anarhichas minor* [64] was higher for diets with less protein and/or more carbohydrates. On the other hand, the protease activity was found to be unresponsive to

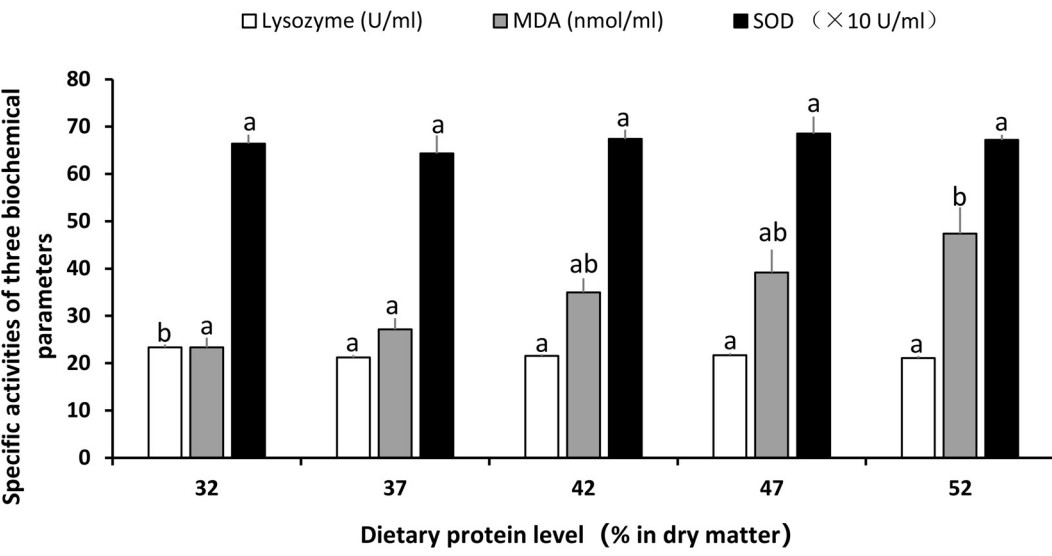

**Fig 3. The three non-specific immune parameters in plasma of *Culter mongolicus* fed with varying dietary protein levels.** Data presented are mean ± SE, different letters on the top of same shape bars means significantly different ($p < 0.05$).

varying dietary protein levels in *Cherax quadricarinatus* [65]. Trypsin activity was not affected by different protein levels diets in *Dicentrarchus labrax* larvae [66]. In the present study, trypsin activity increased with an increase in dietary protein levels. However, trypsin activity began to decrease when the dietary protein levels exceeded 47%. This can be explained by the fact that dietary protein insufficiency results in a reduction in the number amino acids that are responsible for the synthesis and secretion of trypsin and lipase. Carnivorous fish have higher protein requirements than omnivorous and herbivorous fish. Abundant dietary protein stimulates the synthesis and secretion of trypsin. In contrast, excessive protein cannot be fully digested and causes the accumulation of toxic nitrogen in the body, which restrains the activity of trypsin [67, 68]. Various studies have reported that lipase activity was the same across different dietary protein treatments because of the constant lipid content in the diet of some species, such as *Pseudoplatystoma corruscans* [69], *Labeo rohita* fingerlings [61], and *Sardinella brasiliensis* [8]. In the present study, we found that lipase activity showed a similar trend of variability to that of trypsin. This is in line with the result reported in *Puntius gonionotus* fingerlings [70]. Excess protein in the diet can be used for lipid deposition, but not for protein deposition [6, 55]. Excessive dietary protein may convert into massive lipids, thereby restraining the lipase activity. Similarly, an increase in dietary protein can induce an increase in amylase activity in fish [8, 67, 69]. Nevertheless, this is in contrast to the results of a study on *Puntius gonionotus* fingerlings [57]. Moreover, previous studies on *Dentex dentex* [62] and *Labeo rohita* fingerlings [61] indicated that dietary protein did not influence amylase activity. The present results suggest that amylase activity was significantly lower in the 47% protein group, while amylase activity in other treatments did not change. Amylase activity has been reported to decrease with a decrease in dietary carbohydrate content [4, 71]. The content of corn starch in our feed formula increases with decreasing protein to prepare isoenergetic diets. This may be the reason for the reduction in amylase activity when the protein level increased to 47%. On the other hand, according to a study on *Colossoma macropomum* [67], adaptation to the use of carbohydrate sources was also dependent on the lipid concentrations, an increase in dietary lipids reduced the amylase activity. Given the trends in trypsin and lipase activities, we presume that energy is derived from both carbohydrates and lipids. When there were abundant lipids

transformed from proteins, the energy source was ample. Therefore, the demand for carbohydrates decreased, and the activity of amylase receded.

## Immunological effects of dietary protein levels

In a recent study, Li et al. (2018) found that serum lysozyme activity in juvenile sturgeon increased with the increasing dietary protein levels but declined when fish were fed excessive protein [54]. In a similar study on grass carp, the optimal dietary protein levels increased the lysozyme activity, but decreased the MDA concentration [72]. In the present study, the maximum plasma lysozyme activity of *C. mongolicus* appeared in the group fed with 32% protein content, and the maximum MDA concentration observed in the group fed with 52% protein content. Interestingly, the lysozyme activity and the MDA concentration did not vary significantly in the other groups. Organisms have antioxidant defense mechanisms to bate oxidative stress and defend biological systems from free radical toxicity [73, 74]. Excessive dietary protein may have promoted the accumulation of incomplete metabolites and ammonia excretion, leading to increased stress. In addition, we found that SOD activity in this study was not significantly different among the groups with different dietary protein levels. Many factors, such as fish species, feeding duration, age, and environmental factors, could explain the variability in the results of similar studies [75]. Thus, further studies are necessary to clarify the effects of dietary proteins on antioxidant enzymes.

## Conclusion

In summary, this study found that *C. mongolicus* fingerlings reached maximum growth when the dietary protein level was 48.97% to 49.31%. Excessive protein led to retarded growth of *C. mongolicus*, reduced intestinal trypsin and lipase activity, and higher plasma MDA production. Dietary protein levels may have complex effects on the immune system of fish, which requires further research. The present results provide a basis for optimization of artificial culture of *C. mongolicus*, and lay the foundation for further investigation of the nutritional requirements of this species in aquaculture.

## Supporting information

**S1 Data.**
(XLSX)

## Acknowledgments

We are grateful to postgraduate students Mantang Xiong, Zhan Mai, and Puze Wang, from the University of Chinese Academy of Sciences for their assistance for sampling. We appreciate the suggestions from Professor Xiaoming Zhu and Dong Han, and Ms. Yunxia Yang. We also thank the anonymous reviewers of the manuscript for their valuable comments and suggestions.

## Author Contributions

**Conceptualization:** Jing Qian, Jiashou Liu.

**Data curation:** Jing Qian.

**Formal analysis:** Jing Qian, Kai Feng, Wei Li, Jiashou Liu.

**Funding acquisition:** Tanglin Zhang, Jiashou Liu.

**Investigation:** Jing Qian, Lingjun Xiao.

**Methodology:** Jing Qian, Chuansong Liao, Jiashou Liu.

**Project administration:** Jing Qian, Tanglin Zhang, Jiashou Liu.

**Resources:** Jing Qian, Jiashou Liu.

**Software:** Jing Qian.

**Supervision:** Jing Qian, Wei Li, Chuansong Liao, Tanglin Zhang, Jiashou Liu.

**Validation:** Jing Qian, Wei Li, Jiashou Liu.

**Visualization:** Jing Qian, Wei Li, Chuansong Liao, Jiashou Liu.

**Writing – original draft:** Jing Qian.

**Writing – review & editing:** Jing Qian, Lingjun Xiao, Wei Li, Chuansong Liao, Tanglin Zhang, Jiashou Liu.

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
