## [Decision Letter · Decision Letter 0]

30 Dec 2021

PONE-D-21-25983

Effect of dietary protein levels on the growth, enzyme activity, and immunological status of Culter mongolicus fingerlings

PLOS ONE

Dear Dr. Jiashou Liu,

Thank you for submitting your manuscript to PLOS ONE. After careful consideration, we feel that it has merit but does not fully meet PLOS ONE’s publication criteria as it currently stands. Therefore, we invite you to submit a revised version of the manuscript that addresses the points raised during the review process.

We look forward to receiving your revised manuscript.

Kind regards,

Mahmoud A.O. Dawood, PhD

Academic Editor

PLOS ONE

“This work was financially supported by the National R & D Supporting Program (2015BAD25B01), Fund of the State Key Laboratory of Freshwater Ecology and Biotechnology and the earmarked fund for China Agriculture Research System (CARS-45), and the Shandong Provincial Agricultural Seed Engineering Project (No. 2017LZN003).”

“This work was financially supported by the National R & D Supporting Program (2015BAD25B01), Fund of the State Key Laboratory of Freshwater Ecology and Biotechnology and the earmarked fund for China Agriculture Research System (CARS-45), and the Shandong Provincial Agricultural Seed Engineering Project (No. 2017LZN003). We are grateful to postgraduate students Mantang Xiong, Zhan Mai, and Puze Wang, from the University of Chinese Academy of Sciences for their assistance for sampling. We appreciate the suggestions from Professor Xiaoming Zhu and Dong Han, and Ms. Yunxia Yang. We also thank the anonymous reviewers of the manuscript for their valuable comments and suggestions.”

“This work was financially supported by the National R & D Supporting Program (2015BAD25B01), Fund of the State Key Laboratory of Freshwater Ecology and Biotechnology and the earmarked fund for China Agriculture Research System (CARS-45), and the Shandong Provincial Agricultural Seed Engineering Project (No. 2017LZN003).”

Reviewers' comments:

Reviewer's Responses to Questions

**Comments to the Author**

1. Is the manuscript technically sound, and do the data support the conclusions?

Reviewer #1: Partly

Reviewer #2: Yes

2. Has the statistical analysis been performed appropriately and rigorously? 

Reviewer #1: Yes

Reviewer #2: Yes

3. Have the authors made all data underlying the findings in their manuscript fully available?

Reviewer #1: Yes

Reviewer #2: Yes

4. Is the manuscript presented in an intelligible fashion and written in standard English?

Reviewer #1: Yes

Reviewer #2: Yes

5. Review Comments to the Author

Reviewer #1: Dear authors

Your data are new and practical, the manuscript is well-written, and the amout of data should be enough, depending on the journals policies.

Based on these, i would recommend your results for publication.

Reviewer #2: The manuscript investigated the "Effect of dietary protein levels on the growth, enzyme activity, and immunological status of Culter mongolicus fingerlings." The manuscript is well, hence readable. The authors studied a species that appear to new in aquaculture. If it is a new species in aquaculture, this research very relevant as determining nutritional requirement in fish is an important step in the adoption of the fish species in aquaculture. However, before this manuscript is accepted in this prestigious journal minor revision is required.

Keywords: avoid using words in the title of the manuscript.

Line 68: this sentence needs to be specific on what is being referred to: is it fish or something else.

Line 90: cite your earlier research for better reference.

Line 107, what is the basis of the protein level, any reference?

It is not clear in the introduction why protein requirement is being investigated on this particular species. Is it a new species in aquaculture, or old aquaculture species but there is lack of information on its nutritional requirement.

Line 136: ‘’A third of the water was replaced at 19:00 daily. Was’nt this stressful to the fish if we have to take ethics in consideration. Was the rearing system not recirculating. Another issue is the stocking density, don’t the author think 70 fish for this experiment were too many? The use of too many fish for this experiment can be unethical. Explain why you used this number of fish per tank.

Line 137: Although you mentioned that water parameters are presented as average, it is not clear. I suggest you use average/mean with standard deviation or standard error.

Line 183 – Explain the regression formular; what each abbreviation stands for.

Figure 3: The y-axis label needs to be revised. This figure does not represent digestive enzymes but rather non-specific immune parameters/ biochemical parameters. And you may want to change the colour of lysozyme and MDA so that they are clearly distinct.

6. PLOS authors have the option to publish the peer review history of their article (what does this mean?). If published, this will include your full peer review and any attached files.

Reviewer #1: No

Reviewer #2: **Yes: **Ndakalimwe Naftal Gabriel

---

## [Author Response · Author response to Decision Letter 0]

11 Jan 2022

Reviewer #1: Dear authors

Your data are new and practical, the manuscript is well-written, and the amout of data should be enough, depending on the journals policies.

Based on these, i would recommend your results for publication.

Response: Dear reviewer, thanks for your review and approving. Best wishes for you!

Reviewer #2: The manuscript investigated the "Effect of dietary protein levels on the growth, enzyme activity, and immunological status of Culter mongolicus fingerlings." The manuscript is well, hence readable. The authors studied a species that appear to new in aquaculture. If it is a new species in aquaculture, this research very relevant as determining nutritional requirement in fish is an important step in the adoption of the fish species in aquaculture. However, before this manuscript is accepted in this prestigious journal minor revision is required.

Response: Thank you very much for your support to our work. We carefully revised our manuscript according to your valuable comments and suggestions.

Keywords: avoid using words in the title of the manuscript.

Response: Thank you, we changed keywords to: Culter mongolicus, protein requirement, physiological and biochemical indexes

Line 68: this sentence needs to be specific on what is being referred to: is it fish or something else.

Response: It is fish. And the original text was amended to: “It is widely accepted that the level of fish digestive enzyme activity is a useful comparative indicator of food utilization, digestive capacity, and growth performance of the host” (Lines 67-68). 

Line 90: cite your earlier research for better reference.

Response: We added the earlier paper, marked as NO.50, it is in Chinese with English abstract.

Line 107, what is the basis of the protein level, any reference?

Response: Culter mongolicus is a carnivorous species, and according to protein requirement of other carnivorous fish, we set experimental diets (Lines 233-234). 

It is not clear in the introduction why protein requirement is being investigated on this particular species. Is it a new species in aquaculture, or old aquaculture species but there is lack of information on its nutritional requirement.

Response: Culter mongolicus is one new aquaculture species, people feed them with other carnivorous fish’s feed now, and there is lack of information on its nutritional requirement.

Line 136: ‘’A third of the water was replaced at 19:00 daily. Was’nt this stressful to the fish if we have to take ethics in consideration. Was the rearing system not recirculating. Another issue is the stocking density, don’t the author think 70 fish for this experiment were too many? The use of too many fish for this experiment can be unethical. Explain why you used this number of fish per tank.

Response: The new water has been aerated for one day, before slowly flowing into cylinders, so the temperature was the same with old water. With the water changing was slowly, the stress was ignored. And the rearing system was not recirculating. 

The Culter mongolicus fingerlings are small, according to pre-experimentation, 70 fish in one tank was not crowded, and their growth did not stunted.

Line 137: Although you mentioned that water parameters are presented as average, it is not clear. I suggest you use average/mean with standard deviation or standard error.

Response: We use air condition system for temperature control, and the new water has been aerated for one day, before slowly flowing into cylinders, so water parameters are presented as average.

Line 183 – Explain the regression formular; what each abbreviation stands for.

Response: weight-specific growth rate was estimated using broken-line regression analysis (y = L − U × (R − x), y means specific growth rate, x means protein content).

Figure 3: The y-axis label needs to be revised. This figure does not represent digestive enzymes but rather non-specific immune parameters/ biochemical parameters. And you may want to change the colour of lysozyme and MDA so that they are clearly distinct.

Response: Thank you for your suggestion, we revised Fig. 3 accordingly.

---

## [Editor Report · Decision Letter 1]

21 Jan 2022

Effect of dietary protein levels on the growth, enzyme activity, and immunological status of Culter mongolicus fingerlings

PONE-D-21-25983R1

Dear Dr. Jing Qian,

We’re pleased to inform you that your manuscript has been judged scientifically suitable for publication and will be formally accepted for publication once it meets all outstanding technical requirements.

Kind regards,

Mahmoud A.O. Dawood, PhD

Academic Editor

PLOS ONE
---

## [Editor Report · Acceptance letter]

27 Jan 2022

PONE-D-21-25983R1 

Effect of dietary protein levels on the growth, enzyme activity, and immunological status of *Culter mongolicus* fingerlings 

Dear Dr. Liu:

I'm pleased to inform you that your manuscript has been deemed suitable for publication in PLOS ONE. Congratulations! Your manuscript is now with our production department. 

Kind regards, 

on behalf of

Dr. Mahmoud A.O. Dawood 

Academic Editor

PLOS ONE